# Structural basis for the inhibition of RecBCD by Gam and its synergistic antibacterial effect with quinolones

Martin Wilkinson[1], Luca A Troman[2], Wan AK Wan Nur Ismah[3], Yuriy Chaban[1], Matthew B Avison[3], Mark S Dillingham[2]*, Dale B Wigley[1]*

[1]Department of Medicine, Section of Structural Biology, Imperial College London, London, United Kingdom; [2]School of Biochemistry, University of Bristol, Bristol, United Kingdom; [3]School of Cellular and Molecular Medicine, University of Bristol, Bristol, United Kingdom

**Abstract** Our previous paper (Wilkinson *et al*, 2016) used high-resolution cryo-electron microscopy to solve the structure of the *Escherichia coli* RecBCD complex, which acts in both the repair of double-stranded DNA breaks and the degradation of bacteriophage DNA. To counteract the latter activity, bacteriophage λ encodes a small protein inhibitor called Gam that binds to RecBCD and inactivates the complex. Here, we show that Gam inhibits RecBCD by competing at the DNA-binding site. The interaction surface is extensive and involves molecular mimicry of the DNA substrate. We also show that expression of Gam in *E. coli* or *Klebsiella pneumoniae* increases sensitivity to fluoroquinolones; antibacterials that kill cells by inhibiting topoisomerases and inducing double-stranded DNA breaks. Furthermore, fluoroquinolone-resistance in *K. pneumoniae* clinical isolates is reversed by expression of Gam. Together, our data explain the synthetic lethality observed between topoisomerase-induced DNA breaks and the RecBCD gene products, suggesting a new co-antibacterial strategy.

*For correspondence: mark. dillingham@bristol.ac.uk (MSD); d.wigley@imperial.ac.uk (DBW)

**Competing interests:** The authors declare that no competing interests exist.

## Introduction

In bacterial cells, the related AddAB and RecBCD enzymes are helicase-nuclease complexes responsible for initiating homologous recombination from double-stranded DNA breaks (DSBs). This activity underpins many key DNA transactions including DSB repair, phage restriction, and conjugal or transductional recombination (*Dillingham and Kowalczykowski, 2008*; *Wigley, 2013*).

Several observations suggest that selective inhibition of AddAB/RecBCD could be useful in biotechnology and medical applications. For example, they are important for the infectivity and pathogenicity of bacteria that need to resist oxidative and nitrosative attack from neutrophils and macrophages (*Amundsen et al., 2008*; *Darrigo et al., 2016*; *Amundsen et al., 2009*; *Cano et al., 2002*). They protect cells from antibacterials that cause DSBs, such as ciprofloxacin, because they afford a basal level of protection against such damage (*Amundsen et al., 2009*; *Henderson and Kreuzer, 2015*; *González-Soltero et al., 2015*; *Tamae et al., 2008*; *McDaniel et al., 1978*). Moreover, the repair of such DSBs stimulates mutagenesis and recombination via both SOS-dependent and –independent mechanisms which may enhance the acquisition of antibacterial resistance (*López et al., 2007*; *Cirz et al., 2005*). Finally, RecBCD activity interferes with RecET- and Redαβ-mediated 'recombineering', and so efficient bacterial genome engineering may require a *recBC* genetic background (*Court et al., 2002*; *Datta et al., 2008*; *Zhang et al., 1998*; *Muyrers et al., 1999*).

Many phage-encoded proteins manipulate host cell metabolism in order to promote phage replication and lysis (*Liu et al., 2004*). For example, phage λ Gam is a potent inhibitor of the *Escherichia coli* RecBCD complex that helps to protect the phage DNA from degradation (*Sakaki et al., 1973*; *Murphy, 1991*). In this work, we present the structure of Gam bound to RecBCD unveiling an inhibition mechanism based on protein mimicry of a DSB. We also show that *E. coli* and *Klebsiella pneumoniae* cells expressing Gam are hypersensitive to ciprofloxacin. Moreover, inhibition of RecBCD can restore susceptibility to laboratory-selected mutants and clinical isolates of *K. pneumoniae* that are fluoroquinolone resistant. More generally, we argue that the study of other phage-encoded DNA mimics will help to identify novel antibiotic targets and new mechanisms for target inhibition.

## Results

### Gam interacts with the DNA-binding site of RecBCD

The Gam protein exists in two isoforms called GamL and GamS which differ in length (*Sakaki et al., 1973*). Previous work has shown that GamS inhibits RecBCD by competing with DNA binding (*Court et al., 2007*; *Murphy, 2007*). The structure we present here, of the GamS dimer complexed with RecBCD, was determined by cryo-electron microscopy at 3.8 Å resolution (*Figure 1*, *Figure 1—source data 1*, *Figure 1—figure supplements 1* and *2*, *Video 1*). It reveals that the GamS protein does indeed act as a steric block to the binding of DNA (*Figure 1*). The interaction with the duplex DNA-binding 'arm' of the RecB subunit is extensive and overlaps completely with that of the duplex DNA binding site (*Figure 1* and *Video 2*). Furthermore, one of the long N-terminal helical extensions of GamS inserts deeply into RecBCD. It occupies a channel that normally accommodates the nascent 3'-ssDNA tail bound to the RecB helicase subunit, increasing the extent of the steric block (*Figure 1*). Although the structure of the RecBCD complex is closest to that of the initiation complex (*Singleton et al., 2004*) it responds to the binding of Gam by small changes in conformation. The 2B and C-terminal domains of the RecC subunit together with the 2B domain of RecB move as a unit away from the RecB helicase domains. The RecD subunit is also much more flexible.

The crystal structure of the GamS protein alone (*Court et al., 2007*) revealed a pattern of negative charges on the surface that mimicked a DNA duplex, suggesting that molecular mimicry might be involved in binding. Consistent with this idea, a series of acidic side chains on Gam are located in positions equivalent to the phosphates of the DNA backbone and interact with many of the same residues on the RecBCD surface (*Figure 2* and *Figure 2—figure supplement 1*). However, the interaction of RecBCD with GamS is much more extensive than with duplex DNA (*Figure 2A and B*). The interactions of the N-terminal helix with the ssDNA channel are less similar to those involved in ssDNA binding although there is some conservation of hydrophobic contacts across the site (*Figure 2C and D* and *Figure 2—figure supplement 2*). Instead, the main component of the interaction is simply a steric block; the helix makes a snug fit in the channel involving many more contacts than the equivalent contacts with ssDNA (*Video 3*). Overall, the interaction surface involves many contacts and covers approximately 2500 Å$^2$ which explains why the interaction between the proteins is so tight (*Court et al., 2007*; *Murphy, 2007*).

### *Escherichia coli* cells expressing Gam are hypersensitive to ciprofloxacin

Quinolone antibacterials target DNA gyrase and topoisomerase IV and kill cells by stabilising covalent topoisomerase-DNA adducts to produce DSBs. Based on the sensitivity of *recBC* cells to quinolones (*Henderson and Kreuzer, 2015*; *González-Soltero et al., 2015*; *Tamae et al., 2008*; *McDaniel et al., 1978*) and the well-characterised role of RecBCD in the repair of DSBs (*Dillingham and Kowalczykowski, 2008*), we hypothesised that expression of Gam would potentiate the killing effects of ciprofloxacin. To test this hypothesis, we engineered pBAD plasmids to express the two isoforms of Gam (GamL and GamS) from an arabinose-inducible promoter. In the absence of ciprofloxacin, expression of Gam had no apparent effect on the viability of *E. coli* MG1655 (*Figure 3—figure supplement 1*). We next compared the ciprofloxacin minimum inhibitory concentration (MIC) against *E. coli* cells either expressing Gam or containing an empty vector construct. In broth culture, expression of either GamS or GamL reduced the MIC by approximately four-fold compared to the control, and equivalent results were obtained using spot tests on agar plates (*Figure 3A* and *Figure 3—figure supplement 1*). The MIC potentiation effect was dependent on

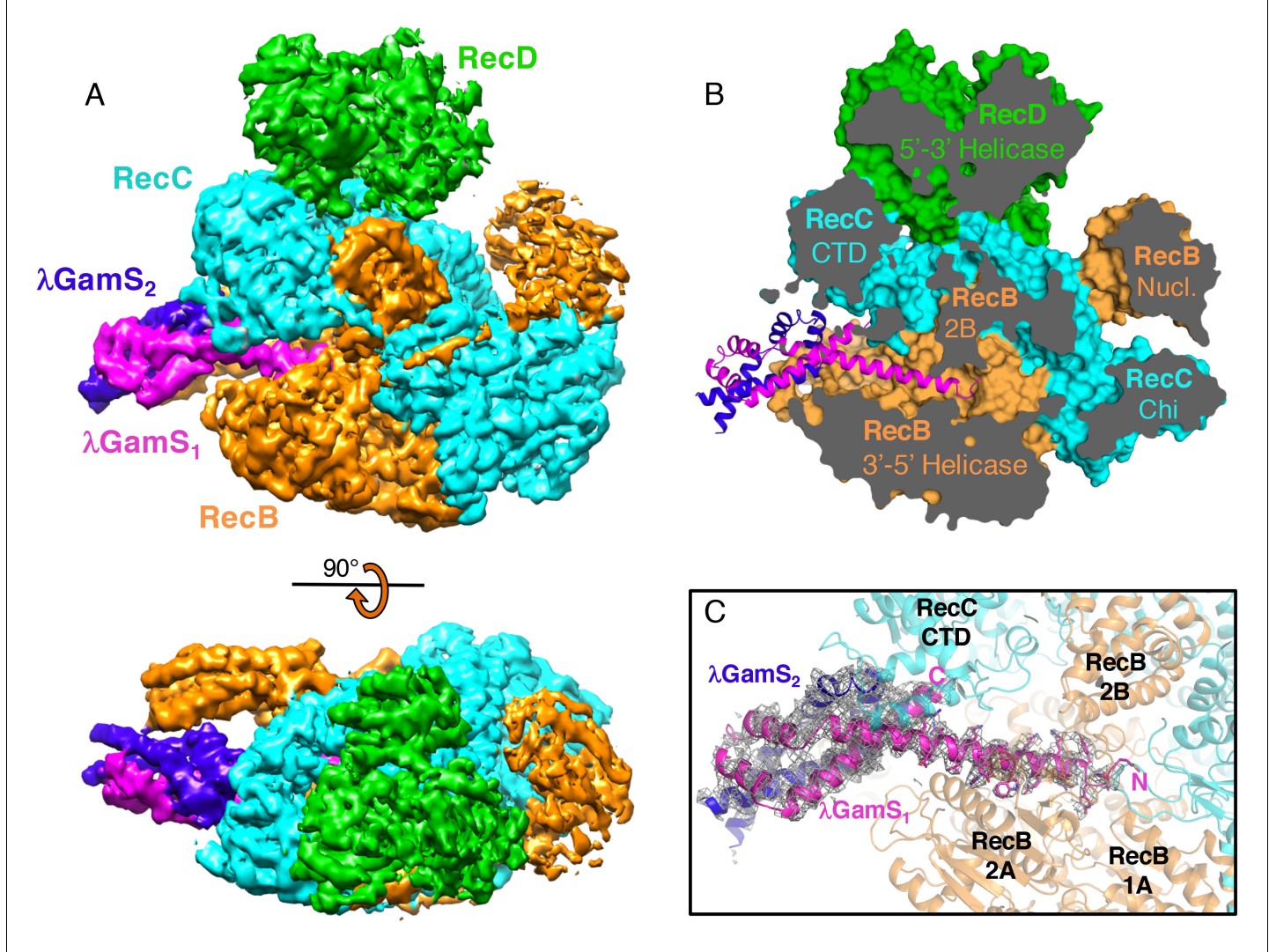

**Figure 1.** The overall structure of the RecBCD/Gam complex. (**A**) Surface representation of the electron density with a ribbon representation of the RecBCD subunits and the GamS protein dimer. (**B**) Cut away of the molecular surface of the RecBCD part of the model with the GamS dimer overlaid showing how the protein enters and fills the channel normally occupied by the 3'ssDNA tail. (**C**) The same view with the electron density for the GamS dimer overlaid.

The following source data and figure supplements are available for figure 1:

**Source data 1.** EM data statistics and Final model.

**Figure supplement 1.** Electron microscopy information.

**Figure supplement 2.** Representation of the local resolution of the GamS$_1$ N-terminal helix for the deposited map compared to the same region in the maps of each of the three sub-classes.

arabinose (*Figure 3—figure supplement 1*) and specific for quinolone-induced DSBs; equivalent experiments measuring the MIC for ampicillin showed no effect of Gam (data not shown). These experiments demonstrate, at the biochemical level, the synthetic lethality observed between topoisomerase malfunction and RecBCD in gene knockout studies (*Tamae et al., 2008*).

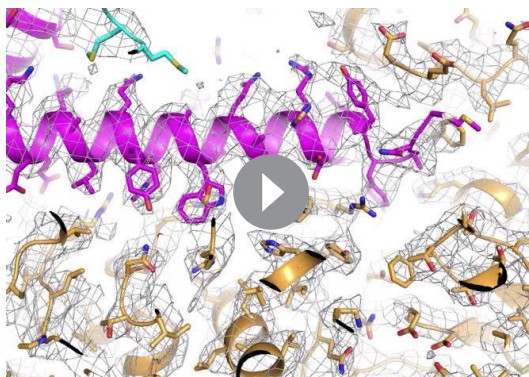

**Video 1.** Electron density map. A region of the electron density is shown in the location of the N-terminal helix of the Gam subunit that is located in the ssDNA binding site of the RecB subunit. Side chain density is clearly visible.

# Inhibition of DSB repair reverses fluoroquinolone resistance in *K. pneumoniae*, including clinical isolates

*K. pneumoniae* is an important opportunistic human pathogen that causes a variety of infections and is of increasing concern due to the recent emergence of antibiotic resistant and hypervirulent strains (*Paczosa and Mecsas, 2016*). To test whether Gam expression increased the susceptibility of wild type *K. pneumoniae* to fluoroquinolones, we used disc susceptibility testing to four different fluoroquinolones. These are standardised assays based on inhibition zone diameter values defined by the CLSI (*CLSI, 2015*) which classify bacterial strains as either resistant, or not resistant, to fluoroquinolones based on the expected efficacy of the drug in a clinical context. Five strains were used: the wild-type strain Ecl8, two laboratory-selected Ecl8 mutant derivatives having sequentially reduced ciprofloxacin susceptibility, and two *K. pneumoniae* clinical isolates. Ecl8, and the single-step reduced susceptibility mutant Ecl8-CIP-M1 are not clinically resistant to any of the four test fluoroquinolones regardless of Gam expression, although cells containing the phage protein are significantly *more* susceptible (i.e. the inhibition zone diameters are greater in all cases) confirming fluoroquinolone potentiation by Gam in this species (*Figure 3B* and *Table 1*). The two-step mutant ECl8-CIP-M2 and clinical isolate R16 are classified as clinically resistant to all four test fluoroquinolones; isolate R20 is resistant to three of them. In all cases, expression of GamL increases fluoroquinolone susceptibility to a level that reverses clinical resistance. Whilst GamS generally had a smaller effect than GamL, it still reversed resistance in all but the least susceptible clinical isolate, R16 (*Table 1*).

## Discussion

The recent work of DuBow and colleagues has elegantly demonstrated the potential for novel antibacterial discovery through the characterisation of bacteriophage:host interactions (*Liu et al., 2004*).

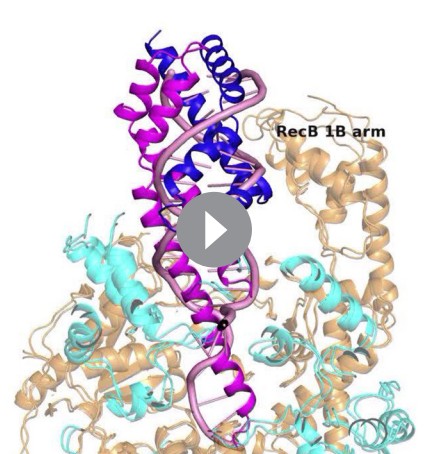

**Video 2.** Overlay of the DNA substrate and the GamS dimer bound to RecBCD.

Indeed, phage have evolved a wide variety of strategies to repurpose host cell functions for their own benefit, including inhibition of key bacterial proteins. These include many examples of proteins that target DNA replication, recombination and repair factors, which often share the primary structure characteristics of DNA mimic proteins (*Wang et al., 2014*). Characterisation of DNA mimic proteins provides an attractive route towards the identification and validation of novel antibiotic targets, because bacterial DNA transactions are not only crucial for survival but are structurally orthogonal to their eukaryotic counterparts (*Robinson et al., 2012*). Moreover, the systematic analysis of the antibiotic sensitivity of single gene knockouts in *E. coli* highlights the potential of targeting DNA binding proteins as part of 'co-antibacterial' strategies to potentiate the effects of existing drugs (*Tamae et al., 2008*).

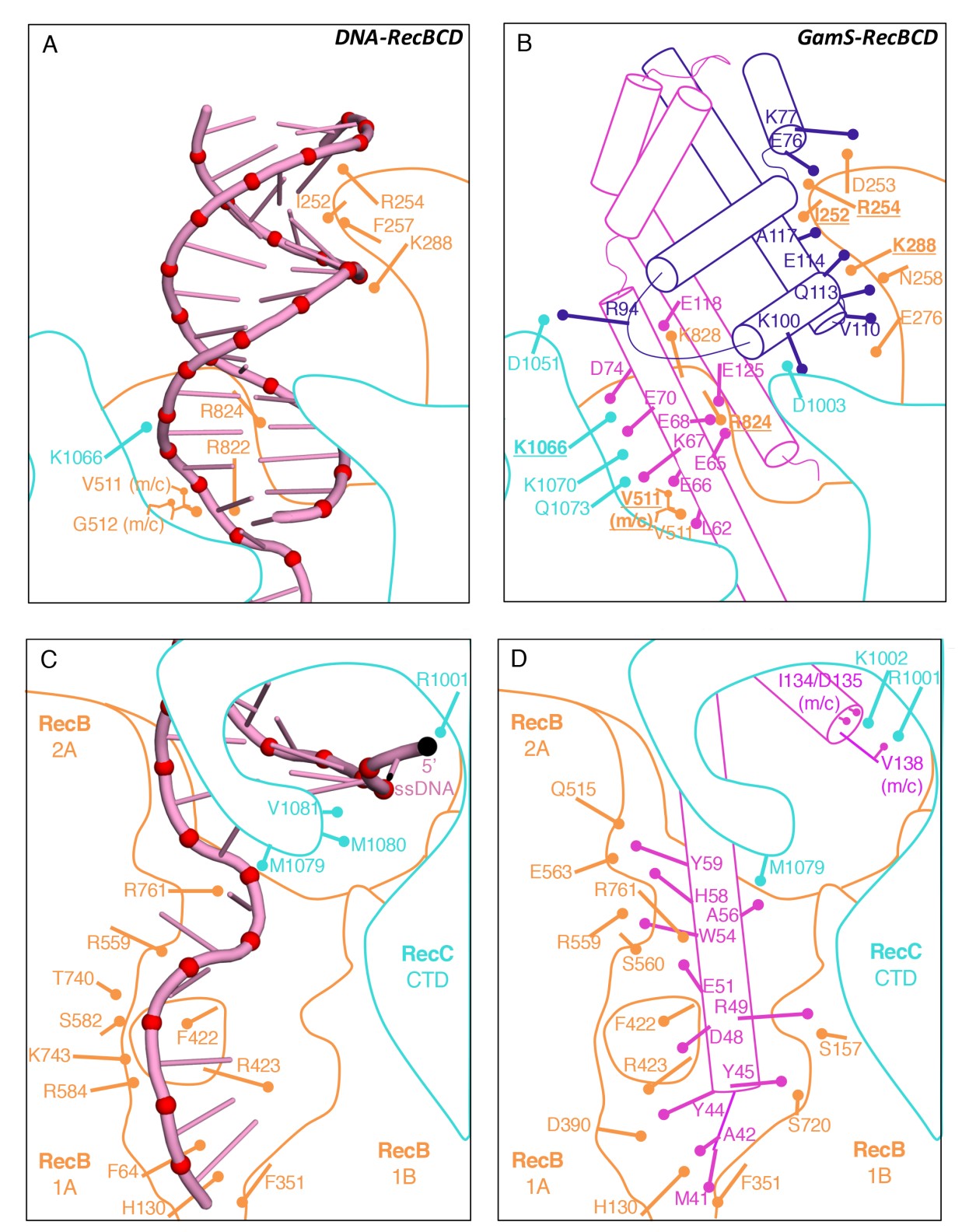

**Figure 2.** Comparison of the RecBCD/Gam and RecBCD/DNA interfaces. (**A**) Cartoon representation of contacts between RecBCD and the duplex portion of bound DNA. RecB is in orange and RecC in cyan in all panels. Interactions involving main chain atoms are denoted as m/c. (**B**) The same interface but with the GamS dimer (shown in magenta and purple). The interaction is much more extensive than with DNA but contacts in common are

*Figure 2 continued on next page*

*Figure 2 continued*

shown in bold. (**C**) Contacts between RecBCD and the ssDNA tails of DNA. (**D**) The same interface but with the GamS dimer. Again the interactions are much more extensive but still include several residues in common (shown in bold).

The following figure supplements are available for figure 2:

**Figure supplement 1.** Space filling representations of the bound DNA substrate and GamS dimer with negative charges coloured in red.

**Figure supplement 2.** Aspects of molecular mimicry are shown in a comparison of the interactions in RecBCD/DNA complex with that of the RecBCD/Gam complex in the region of the 3'-tail of the DNA substrate.

In this work, we determined the structural basis for inhibition of the RecBCD complex by the phage-encoded DNA mimic Gam, and exploited this interaction to show that inhibition of host cell DSB repair is a useful co-antibacterial strategy alongside quinolone drugs. GamL reversed resistance to all four tested fluoroquinolones in clinical isolates of multi-drug resistant *K. pneumoniae*, a human pathogen, as defined by an assay validated to predict clinical efficacy of antibacterial drugs (*CLSI, 2015*). Our data show that the Gam protein works by blocking the DNA binding site, partly by imitating both the single- and double-stranded portions of broken DNA. Gam has also been reported to interact with the SbcCD complex, another nuclease implicated in DNA repair (*Kulkarni and Stahl, 1989*). However, inhibition of SbcCD is unlikely to contribute to the antibacterial potentiation effect observed here, because cells lacking SbcCD activity are not sensitive to ciprofloxacin (*Henderson and Kreuzer, 2015*; *Aedo and Tse-Dinh, 2013*; *Liu et al., 2010*). DNA mimicry has been observed previously for phage-encoded proteins that target type I restriction endonucleases and glycosylases (*Kennaway et al., 2009*; *Baños-Sanz et al., 2013*; *Cole et al., 2013*) and it may be a common mechanism for bacteriophages to modulate DNA replication and repair in their hosts. Bacteriophage P22 codes for another, distinctive RecBCD inhibitor called Abc2 but this operates by a poorly-characterised mechanism (*Murphy, 2000*). Moreover, a small molecule inhibitor of AddAB/RecBCD (ML328) has recently been developed (*Bannister et al., 2010*). It will be of great interest to understand the inhibitory mechanisms of these molecules, expanding our understanding of how we can control bacterial DNA repair and, potentially, exploit this knowledge to combat antibacterial resistance.

## Materials and methods

### Cloning of the bacteriophage λ Gam protein

For over-expression in *E. coli*, the gene encoding bacteriophage lambda GamS, corresponding to residues 41–138 of the full-length Gam protein, was synthesised with codon optimisation for *E.coli* expression (GeneArt, Thermo Fisher, Waltham, MA, USA). GamS was cloned into pET22b, between NdeI and BamHI, using In-Fusion cloning. For fluoroquinolone potentiation studies, the genes encoding GamS and full length GamL were cloned into the pBAD322K plasmid (which replicates in both *E. coli* and *K. pneumoniae*) under the control of an arabinose-inducible promoter (a gift from Prof. John Cronan, University of Illinois [*Cronan, 2006*]) using the NcoI and HindIII restriction sites. All plasmid sequences were verified by direct DNA sequencing.

### Expression and purification of the RecBCD-Gam complex

GamS protein was over-expressed using the pET22b construct in BL21 (DE3) cells with 4 hr induction at 27°C. The purification procedure for

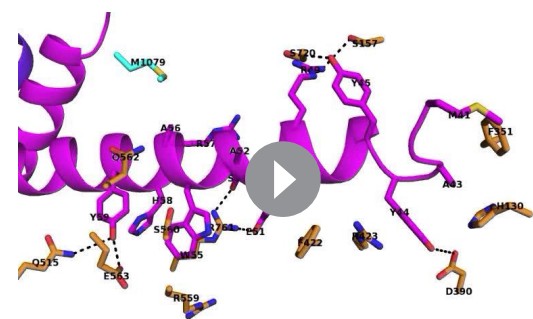

**Video 3.** Details of the interactions between the N-terminal helix of Gam and the RecB subunit.

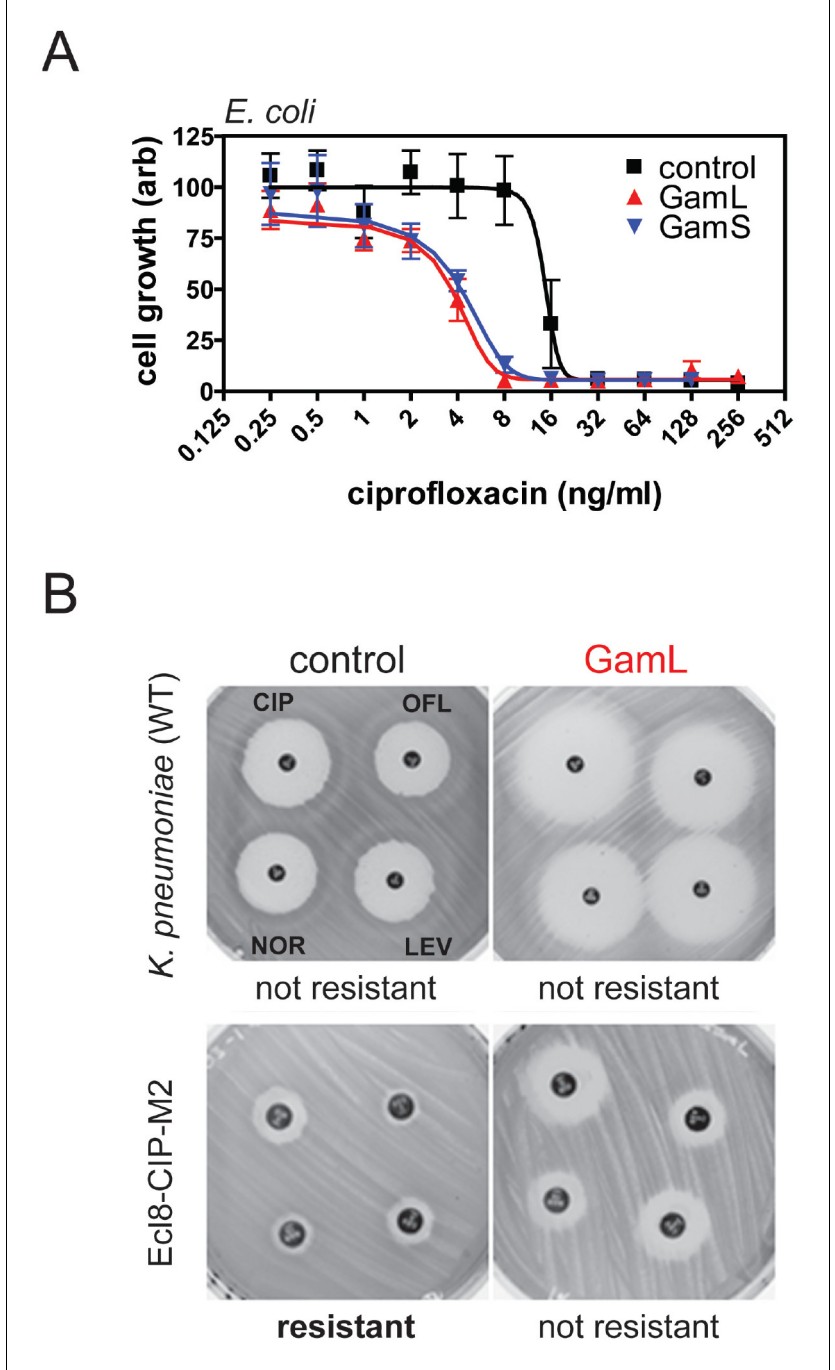

**Figure 3.** Inhibition of bacterial DSB repair potentiates fluoroquinolone antibiotics. (**A**) Ciprofloxacin minimum inhibitory concentration (MIC) assay against *E. coli* MG1655 cells in the presence or absence of Gam isoforms as indicated. Experiments were performed as described in the Materials and methods in the presence of arabinose to induce expression of the small or large isoforms of Gam. Control experiments were performed under identical conditions with the empty pBADK expression vector. (**B**) Disc susceptibility assays are standardised tests that quantify antibacterial susceptibility in terms of an inhibition zone diameter, and also classify bacterial strains as either resistant, or not resistant, to antibacterial agents based on expected drug efficacy in a clinical setting. The experiments were performed and interpreted in accordance with CLSI guidelines (**CLSI, 2015**) with a range of fluoroquinolones (CIP; ciprofloxacin, OFL; ofloxacin, NOR; norfloxacin, LEV, levoxofloxacin). Illustrative results for wild type *K. pneumoniae* (Ecl8) and a resistant strain (Ecl8-CIP-M2) are shown using the GamL isoform. Note that the magnification of the plates is different for the two strains as is apparent from the disc sizes. The inhibition zone diameters for all strains, with both GamL and GamS, are summarised in *Table 1*.
*Figure 3 continued on next page*

*Figure 3 continued*
The following figure supplement is available for figure 3:

**Figure supplement 1.** Spot tests for ciprofloxacin sensitivity on agar plates.

GamS was as described (*Court et al., 2007*). As observed previously, GamS purifies as a dimer. *E. coli* RecBCD complex was expressed and purified as described (*Wilkinson et al., 2016*). A 2:1 molar excess of dimeric GamS was incubated with RecBCD for 15 min at room temperature prior to size-exclusion chromatography with a Superdex 75 10/300 GL column, to separate free GamS. The buffer used contained: 25 mM Tris pH 7.5, 50 mM NaCl and 0.5 mM TCEP (pH 7.0). The peak fractions were pooled and concentrated to 1.4 mg/ml using Vivaspin 30 KDa molecular weight cut-off centrifugal concentrators.

## Cryo-electron microscopy grid preparation and data collection

Thinned C-flat 1 × 1 μm holey carbon film grids were prepared as described (*Wilkinson et al., 2016*). Grids were thinned using a total of 3 min of glow discharge (in 30 s steps), left for two weeks and then treated using a 1 mM solution of Amphipol A8-35. Sample (3 μL) was evenly applied to the grid before blotting and freezing in liquid ethane using a Vitrobot Mark IV (FEI). For blotting, a relative blot force of −4 for 1 s at 4°C was used. Data were collected using a Titan Krios I microscope

**Table 1.** Disc susceptibility assay for *K. pneumoniae* Ecl8wt, derived mutants and clinical isolates expressing GamL or GamS from pBADK. The disc susceptibility assay were performed according to the CLSI protocol (*CLSI, 2015*) using Mueller-Hinton agar with 0.2% (w/v) arabinose to stimulate expression of cloned genes (in bracket) and 30 mg/L kanamycin to select for the pBADK plasmids. Resistance breakpoints are as set by the CLSI (*CLSI, 2015*). Values shaded designate resistance. Values reported are the means of three repetitions rounded to the nearest integer.

| *K. pneumoniae* strains | Diameter of growth inhibition zone (mm) around fluoroquinolone disc | | | |
|---|---|---|---|---|
| | Fluoroquinolone (µg in disc) | | | |
| | Ciprofloxacin (5) | Levofloxacin (5) | Norfloxacin (10) | Ofloxacin (5) |
| Ecl8wt pBADK (Control) | 34 | 32 | 30 | 30 |
| Ecl8wt pBADK (GamL) | 40 | 37 | 37 | 37 |
| Ecl8wt pBADK (GamS) | 46 | 42 | 41 | 39 |
| | | | | |
| Ecl8-CIP- M1 pBADK (Control) | 24 | 21 | 20 | 19 |
| Ecl8-CIP-M1 pBADK (GamL) | 31 | 27 | 24 | 24 |
| Ecl8-CIP-M1 pBADK (GamS) | 28 | 27 | 25 | 25 |
| | | | | |
| Ecl8-CIP-M2 pBADK (Control) | 14 | 12 | 11 | 8 |
| Ecl8-CIP-M2 pBADK (GamL) | 21 | 17 | 16 | 14 |
| Ecl8-CIP-M2 pBADK (GamS) | 21 | 18 | 16 | 14 |
| | | | | |
| R16 pBADK (Control) | 8 | 8 | 8 | 6 |
| R16 pBADK (GamL) | 17 | 16 | 16 | 16 |
| R16 pBADK (GamS) | 15 | 15 | 15 | 12 |
| | | | | |
| R20 pBADK (Control) | 15 | 13 | 14 | 11 |
| R20 pBADK (GamL) | 20 | 19 | 19 | 15 |
| R20 pBADK (GamS) | 18 | 17 | 17 | 15 |

operated at 300 KV at eBIC, Diamond, UK. Zero loss energy images were collected automatically using EPU (FEI) on a Gatan K2-Summit detector in counting mode with a pixel size corresponding to 1.34 Å at the specimen. A total of 358 images were collected with a nominal defocus range of −1.2 to −2.4 µm. Each image consisted of a movie stack of 25 frames with a total dose of 36 electrons/Å$^2$ over 10 s corresponding to a dose rate of 6.5 electrons/pixel/s.

## Data processing

All 25 movie frames for each image stack were aligned and summed using Motioncorr (*Li et al., 2013*) prior to processing with Relion1.4 (*Scheres, 2016*). The actual defocus and other contrast transfer function (CTF) parameters for each summed movie stack were determined using Gctf (*Zhang, 2016*). Outlying micrographs were removed based on a number of criteria, leaving 334 images with Thon rings extending to an estimated resolution range from 6.2–2.8 Å (mean of 3.4 Å) and defocus range between −0.6 to −2.9 µm. The program Gautomatch (*Urnavicius et al., 2015*) was used for automated, template-free particle picking with a circular particle diameter of 140 Å, picking 134,124 particles. Initially, reference-free 2D classification was used to remove poor particles from autopicking, leaving 122,796 particles judged to represent potential protein complexes. These were subjected to 3D classification using the RecBCD:DNA crystal structure (PDB:1 W36) (*Singleton et al., 2004*), low-pass filtered to 45 Å, as a starting model. All four classes generated contained complexes of GamS with RecBCD although with different occupancy of the RecB nuclease and RecD 2A/2B domains. All particles were initially kept together for more robust classification after correcting for single-particle movement and per-frame radiation damage in particle polishing. 3D refinement of the 122,896 particles yielded a map with a resolution of 4.4 Å before masking (as esti-mated by the 'gold-standard' Fourier Shell Correlation (FSC) at the 0.143 cut-off criterion). After application of an auto-generated mask in Relion, the resolution of the map was 4.0 Å. Particle polish-ing improved the masked resolution to 3.8 Å.

At this point 3D classification without alignment (*Scheres, 2016*) was used to separate the differ-ent conformations within the data. The data were split into 10 classes with the majority (82%) falling into three major distinct classes. There were 30% of the particles in a class that lacked density for the RecB nuclease. The other two classes both represented the full complex with a similar spread of views but, when refined separately, showed slightly different conformations of a block of domains including the RecB 2B domain, part of the RecC CTD and the RecD 2A/2B domains. Since the GamS and surrounding structure was consistent for all three classes, they were summed for the final map at 3.8 Å. The final applied b-factor for sharpening used in post-processing was −70 Å$^2$ with the resulting FSC plot shown in *Figure 1—figure supplement 1*.

## Model building and refinement

Scripts for map conversion, cell matching and refinement in Refmac were kindly provided by Garib Murshudov (MRC-LMB). The recent cryo-EM structure of RecBCD in complex with DNA (PDB: 5LD2) (*Wilkinson et al., 2016*), and the crystal structure of GamS (PDB: 2UUZ) (*Court et al., 2007*), were used as starting models for global docking in Chimera (*Pettersen et al., 2004*). Jelly-body Refmac refinement was used to correct for conformational changes in the model from the template struc-tures. The entire model was carefully edited and fit using real-space refinement in Coot (*Emsley et al., 2010*) with occasional jelly-body refinement with Refmac, monitoring geometry statis-tics throughout. Near the end, phenix.real_space_refine (*Afonine et al., 2013*) was used to generate the final model, assign two group B-factors per residue and model statistics (see *Figure 1—source data 1*).

## Bacterial strains and fluoroquinolone sensitivity tests

Fluoroquinolone sensitivity was tested in *E. coli* MG1655 and five different strains of *K. pneumoniae*; wild-type isolate Ecl8 (a fully susceptible strain (*Forage and Lin, 1982*) and a gift from Dr T. Schneiders, University of Edinburgh); Ecl8-CIP-M1, a mutant derivative selected for growth on LB agar containing 30 ng/mL of ciprofloxacin following plating of 100 µL of an overnight culture of Ecl8; Ecl8-CIP-M2, a mutant derivative selected for growth on LB agar containing 4 µg/mL of ciprofloxacin following plating of 100 µL of an overnight culture of Ecl8-CIP-M1; clinical isolates R16 and R20 selected because of their reduced ciprofloxacin susceptibility (cultured from the blood of patients

being treated at Southmead Hospital, Bristol and a gift of Prof A. P. MacGowan). Minimum inhibitory concentration (MIC) assays were performed as described previously with some modifications (*Wiegand et al., 2008*). Ciprofloxacin stocks of 4 μg/ml were made by dissolving in 0.1M HCl. Antibiotic serial dilutions were made in Mueller Hinton broth (MHB). For assays using *E. coli,* cells were grown in the absence of the inducer arabinose at 37°C to reach stationary phase. Cells were then diluted in MHB to $OD_{600}$ ~$10^{-4}$ or an approximate cell count of $10^5$ $cfu.ml^{-1}$ within each well of a 96 well plate. The plates were incubated at 37°C for 18 hr. Where appropriate, the wells were supplemented with kanamycin (50 μg/ml), arabinose (1% w/v), and various antibiotics at the indicated concentrations. For *K. pneumoniae,* antibiotic susceptibility disc tests followed the CLSI protocol (*CLSI, 2015*). Media were supplemented with kanamycin (30 μg/ml) and arabinose (0.2% w/v) as appropriate. The zone of growth inhibition around each disc was determined, in triplicate, by taking multiple measurements of the diameter and quoting the mean value, rounded to the nearest integer.

## Acknowledgements

WAKWNI was funded by a postgraduate scholarship from the Malaysian Ministry of Education. We thank Alistair Siebert, Daniel Clare and Sonja Welsch at the EBIC EM facility at Diamond for help with EM data collection. Protein coordinates been deposited at the protein databank (ID code 5MBV) and the electron density map has been deposited at the EM databank (ID code 3460).

## Additional information

### Funding

| Funder | Grant reference number | Author |
| --- | --- | --- |
| Wellcome | 095519/B/11/Z | Dale B Wigley |
| Medical Research Council | MR/N009258/1 | Dale B Wigley |
| Cancer Research UK | A12799 | Dale B Wigley |
| Wellcome | 100401/Z/12/Z | Mark S Dillingham |
| Engineering and Physical Sciences Research Council | EP/M027546/1 | Matthew B Avison |

The funders had no role in study design, data collection and interpretation, or the decision to submit the work for publication.

### Author contributions

MW, LAT, WAKWNI, Conception and design, Acquisition of data, Analysis and interpretation of data, Drafting or revising the article; YC, Acquisition of data; MBA, MSD, DBW, Conception and design, Analysis and interpretation of data, Drafting or revising the article

### Author ORCIDs

Mark S Dillingham, http://orcid.org/0000-0002-4612-7141
Dale B Wigley, http://orcid.org/0000-0002-0786-6726

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
