## [Decision Letter]

Thank you for submitting your article "Structural basis for the inhibition of RecBCD by Gam and its synergistic antibacterial effect with quinolones" for consideration by *eLife*. Your article has been favorably evaluated by Richard Losick (Senior Editor) and two reviewers, one of whom is a member of our Board of Reviewing Editors. The reviewers have opted to remain anonymous.

The reviewers have discussed the reviews with one another and the Reviewing Editor has drafted this decision to help you prepare a revised submission. Your covering letter should state how you have responded to the reviewer comments.

The manuscript extends the recent structure of RecBCD-DNA from the same group by presenting a new structure [single-particle cryoEM] of RecBCD complexed with the phage λ Gam protein that blocks access of DNA to RecBCD by being a DNA mimetic, thereby inhibiting DNA degradation by RecBCD. One might have guessed this a priori but nevertheless it is a nice structure. Importantly, the authors then go on to show that presence of Gam potentiates the effects of quinolone antibiotics in *E. coli* and a pathogenic *Klebsiella*, thereby showing potential for new antibacterials. Taken together a neat story; the paper is easy to read and follow, though the writing could be improved (see below), and the conclusions are supported by the data.

Specific comments:

1) The whole manuscript feels as if written in a hurry and is somewhat 'ragged at the edges'. It would benefit from careful editing.

2) The authors should more clearly describe the difference (in amino acid sequence and other biochemical properties) between GamL and GamS. Do the authors have any idea why GamL has a stronger effect in vivo than GamS? It is a little bit non-ideal that the structural study used GamS instead of GamL given the stronger effect of GamL in vivo. Also, why did the authors choose to use the N-terminally truncated form of GamS (41-138)? Presumably removal of those residues might be required for crystallization, but probably not for cryoEM, so it would have been better to use the full-length protein?

3) In the subsection “*Escherichia coli* cells expressing Gam are hypersensitive to ciprofloxacin” and later [subsection “Cloning of the bacteriophage λ Gam protein”]. pBAD plasmids are based on pBR322 and are *not* broad host range/specificity.

4) Figure 1 legend: 'major class' – not clear; in Figure 1—figure supplement 2 there are classes of 30%, 30% and 22%.

5) Introduction, second paragraph: The protection must be because of the repair role of RecBCD. Why not state this explicitly?

6) Discussion, last paragraph: Delete 'key' – it adds nothing.

7) Figure 1 legend: Replace 'which' by 'with'.

8) Figure 3. The designations 'not resistant' or resistant seem a little over simplistic? E.g. in the bottom right panel 'less resistant' would seem a more apt description?

9) Table 1 'from' not 'on' pBADK.

10) Figure 3—figure supplement 1. Almost illegible and 'offending' typescript top left corner of top panel.

11) It is worth mentioning in the Introduction, that Gam has also been shown to inhibit other nucleases in the cell. The authors should also consider in their analysis of the bacteria cell growth experiments whether any of the strains express these other nucleases, and whether it is possible some of the growth inhibition could result from these other activities.

---

## [Author Response]

*Specific comments:*

*1) The whole manuscript feels as if written in a hurry and is somewhat 'ragged at the edges'. It would benefit from careful editing.*

We have tried to smooth out some of the rough edges in the revised version, not least of all by answering the reviewers’ comments as below.

*2) The authors should more clearly describe the difference (in amino acid sequence and other biochemical properties) between GamL and GamS. Do the authors have any idea why GamL has a stronger effect in vivo than GamS? It is a little bit non-ideal that the structural study used GamS instead of GamL given the stronger effect of GamL in vivo. Also, why did the authors choose to use the N-terminally truncated form of GamS (41-138)? Presumably removal of those residues might be required for crystallization, but probably not for cryoEM, so it would have been better to use the full-length protein?*

The reviewers are confused about the relationship between GamS and GamL. GamS is a naturally occurring variant of GamL that is truncated at the N-terminus. It is actually rather difficult to make homogenous GamL because it gets truncated naturally during expression so is a mixture of GamS and GamL. In fairness, we do not know if this also applies to the in vivo experiments or even if the higher efficacy of GamL in vivo is due to higher expression levels or stability. Although, the reviewers are correct that the issues of a longer protein for crystallisation do not apply, a heterogeneous mixture is still not ideal for cryoEM. We therefore chose to use the GamS variant as we knew this would be homogeneous at the protein level as this still acts as an effective inhibitor. It may be that engineering the N-terminus to prevent proteolysis, or indeed increase efficacy of inhibition, would be effective but that is a whole project in itself.

*3) In the subsection “Escherichia coli cells expressing Gam are hypersensitive to ciprofloxacin” and later [subsection “Cloning of the bacteriophage λ Gam protein”]. pBAD plasmids are based on pBR322 and are not broad host range/specificity.*

The referees are correct to point out that pBADK is not a truly broad host range plasmid. However, the origin will replicate in many members of the family *Enterobacteriaceae*, including *K. pneumoniae* and *E. coli* that are used in this work. We have modified the text by removing the words “broad specificity” (Results) and “broad host range” (Methods). Instead we now simply point out in the Methods that pBADK can replicate in both of the organisms studied here.

*4) Figure 1 legend: 'major class' – not clear; in Figure 1—figure supplement 2 there are classes of 30%, 30% and 22%.*

“Major class of particles” now deleted from the legend.

*5) Introduction, second paragraph: The protection must be because of the repair role of RecBCD. Why not state this explicitly?*

As requested, we now state that protection against quinolones is due to DNA repair.

*6) Discussion, last paragraph: Delete 'key' – it adds nothing.*

The word “key” is now deleted.

*7) Figure 1 legend: Replace 'which' by 'with'.*

The word “which” replaced by “with”.

*8) Figure 3. The designations 'not resistant' or resistant seem a little over simplistic? E.g. in the bottom right panel 'less resistant' would seem a more apt description?*

We understand the point the referees are making. It is of course the case that bacterial strains display a continuum of susceptibility levels to antibacterial agents which are condition-dependent, and the effect of Gam is to make all the strains we study more susceptible to quinolones regardless of whether they are classified as clinically resistant or not. However, in the context of the standardised disc susceptibility assays used here, the terms "resistant" and "not resistant" have a specific meaning related to clinical definitions of resistance in *Enterobacteriaceae* isolates. This is a binary definition based on empirical data from the US Clinical Laboratory Standards Institute that is used by clinicians to make antibiotic prescribing decisions, and the consensus in the AMR field is that the terms "more resistant" and "less resistant" should be avoided. The significance for our work is that, in principle, inhibition of DNA break repair in *K. pneumoniae* can restore the ability of fluoroquinolones to treat infections that would otherwise fail to respond to antibacterials. We have rewritten the relevant part of the Results section and the legend to Figure 3 to clarify these points.

*9) Table 1 'from' not 'on' pBADK.*

The word “on” replaced by “from” in Table 1.

*10) Figure 3—figure supplement 1. Almost illegible and 'offending' typescript top left corner of top panel.*

Unfortunately, without “photoshopping” the image we can’t get rid of the plate labelling or edge of the plates in the corner of the figures. However, this does at least show these are “real” data! However, we note that the abbreviation “cfu” was not defined and was a little small in the figure. Consequently, we have increased the font size and defined the term in the figure legend.

*11) It is worth mentioning in the Introduction, that Gam has also been shown to inhibit other nucleases in the cell. The authors should also consider in their analysis of the bacteria cell growth experiments whether any of the strains express these other nucleases, and whether it is possible some of the growth inhibition could result from these other activities.*

We thank the referees for this comment which we now directly address in the Discussion. We assume the nuclease in question is the SbcCD complex, which has been reported to be another interaction target of Gam. Inhibition of SbcCD is probably not important in the ciprofloxacin potentiation observed here because, somewhat surprisingly, at least three independent studies show that cells lacking SbcCD activity are not sensitive to ciprofloxacin.